

**Karst aquifer discharge response to rainfall interpreted as anomalous transport**
Dan Elhanati[1], Nadine Goeppert[2, 3], Brian Berkowitz[1]
[1]Department of Earth and Planetary Sciences, Weizmann Institute of Science, Rehovot, Israel
[2]Institute of Applied Geosciences, Division of Hydrogeology, Karlsruhe Institute of
Technology (KIT), Karlsruhe, Germany
[3]Institute of Geological Sciences, Hydrogeology, Free University Berlin, Germany
Correspondence to: Dan Elhanati (dan.elhanati@gmail.com)



## Abstract

The discharge measured in karst springs is known to exhibit distinctive long tails during recession times following distinct discharge peaks of short duration. The long-tail behavior is generally attributed to the occurrence of tortuous, ramified flow paths that develop in the underground structure of karst systems. Modeling the discharge behavior poses unique difficulties because of the poorly-delineated flow path geometry and generally scarce information on the hydraulic properties of catchment-scale systems. In a different context, modeling of long-tailed behavior has been addressed in studies of chemical transport. Here, an adaptation of a continuous time random walk – particle tracking (CTRW-PT) framework for anomalous transport is proposed, which offers a robust means to quantify long-tailed breakthrough curves that often arise during chemical species transport under various flow scenarios. A theoretical analogy is first established between partially water-saturated karst flow, characterized by temporally varying water storage, and chemical transport involving accumulation and release of a chemical tracer. This analogy is then used to develop and implement a CTRW-PT model. Application of this numerical model to examination of three years of summer rainfall and discharge data from a karst aquifer system – the Disnergschroef high alpine site in the Austrian Alps – is shown to yield robust fits between modeled and measured discharge values. In particular, the analysis underscores the predominance of slow diffusive flow over rapid conduit flow. The study affirms the analogy between partially saturated karst flow and chemical transport, exemplifying the compatibility of the CTRW-PT model for this purpose. Within the specific context of the Disnergschroef karst system, these findings highlight the predominance of slow diffusive flow over rapid conduit flow. The agreement between measured and simulated data supports the proposed analogy between partially saturated karst flow and chemical transport; it also highlights the potential ability of the anomalous transport framework to further enhance modeling of flow and transport in karst systems.



## 1 Introduction

Aquifers consist of various geological formations through which water can flow and carry chemical species. The abundance of structural heterogeneities, ranging from intricate grain arrangements at the pore scale to larger geologic structures and discontinuities at the meso- and macroscopic scales, introduces irregular and tortuous flow paths that cannot be delineated without a full physical description of the system. Achieving an accurate representation of flow and transport therefore becomes increasingly difficult with an increase in the scale and complexity of the groundwater system.

Karst systems, in particular, are known as structurally complex aquifers. They are composed of many interconnected conduits, fractures and voids formed through the dissolution of soluble rocks like limestone, dolostone and gypsum, which leads to the occurrence of multiple and ramified flow paths (Bakalowicz, 2005). Karst terrains are usually described, in a vertical cross-section, by distinct hydrological layers whose structure affect the response of the system to incoming precipitation: (1) the surface soil layer; (2) the interface between the soil layer and the deeper saturated zone (epikarst); and (3) the deep underground, mostly phreatic, zone (endokarst). The soil and epikarst layers, known collectively as the exokarst, are known to exhibit lateral flow of water above and below ground, until water reaches fractures or conduits that allow them to flow rapidly to the endokarst. This allows for some water to infiltrate downwards, while some may remain in the vadose zone and be subjected to both percolation and evapotranspiration (Jukić and Denić-Jukić, 2009). The epikarst and endokarst layers each consist of a primary (matrix) porosity composed of all bulk pores, a secondary porosity composed of the smaller joints and fissure developed during diagenesis and/or by tectonic processes, and a tertiary porosity of large fractures and voids (conduits) created due to karstification (Ford and Williams, 2007). The different types of porosities usually exhibit distinct flow patterns: rapid flow in the conduits and slow diffusive flow in the smaller fissures and the matrix. The different karst layers may exhibit a changing role in facilitating the flow or retention of water through the system as a function of water level or recharge intensity (Hartmann et al., 2014).

To date, various hydrological models have been developed specifically for karst systems, to describe the commonly observed flow and transport patterns that are specific to karst systems. In particular, distributed models rely on creating a grid of cells with different hydrological parameters (e.g., Anderson & Radić, 2022; Chen et al., 2017; Kaufmann & Turk, 2016), while



lumped parameter models parameterize the characteristics of the system. Lumped parameter
models are based on different system conceptualizations (e.g., Chen and Goldscheider, 2014;
Cinkus et al., 2023b; Fleury et al., 2009; Jukić and Denić-Jukić, 2009; Mazzilli et al., 2019;
Rimmer and Salingar, 2006; Tritz et al., 2011), as well as neural network approaches (e.g.,
Afzaal et al., 2020; Cinkus et al., 2023b; Kratzert et al., 2018; Renard and Bertrand, 2017;
Wunsch et al., 2022). A common, significant feature encountered in karst systems – which is
difficult to capture in models – is the interplay of rapid and slow flow which manifests as long-
tailed measurements of both discharge rates (e.g., Frank et al., 2021) and chemical tracer
concentrations (e.g., Goeppert et al., 2020) observed at karst springs.
In many systems that exhibit highly variable spatial velocity distributions or temporal
behaviors, measurements of long tails in arrival times may be encountered. In the context of
chemical transport in porous media, long tails in the arrival time of chemical tracers have long
been a subject of study. Anomalous transport, which describes chemical transport that deviates
from the behavior described by the traditional Advection-Dispersion Equation (ADE), is
prevalent in many system and transport scenarios (Berkowitz et al., 2006); deviations from
solutions of traditional transport equations were observed even for non-dispersive diffusion
(Cortis and Knudby, 2006). It has been shown that higher subsurface heterogeneity increases
the degree of anomalous transport by inducing longer than expected (for Fickian transport)
arrival times (Edery et al., 2016, 2014). Traditional ADE based models, which rely on
averaging the physical traits of the medium into a single coefficient, do not accurately predict
transport in many cases. To correctly describe long-tailed events, various modeling approaches
have been developed. Among these, the Continuous Time Random Walk (CTRW) framework
has emerged as suitable for simulating diverse transport scenarios, including the behavior of a
long-time field-scale hydrological catchment (Dentz et al., 2023). The CTRW framework
accounts for anomalous transport behavior and offers a more physically realistic representation
of the transport processes that are encountered in real-world groundwater systems. The
framework defines waiting time and step length distributions that are applied in random walks
which are continuous in time, thereby capturing the complexity of transport processes
(Berkowitz et al., 2006).
In the current study, the CTRW framework, which has been developed to model anomalous
chemical transport, is utilized to quantify long-tailing of water flow in karst systems. In this
context, data from the Disnergschroef alpine study site in Vorarlberg, Austria are revisited
(Frank et al., 2021). This high-alpine karst system has been thoroughly studied and offers a





well-defined spatial catchment. The known extent of its recharge basin and the corresponding
single spring which serves as its outlet allow for measurements of both recharge and discharge.
Previous studies (e.g., Frank et al., 2021) identified a distinct discharge response approximately
5.5 hours after a rainfall event, with variations in electrical conductivity, indicative of fresh
rainfall arriving at the spring outlet, observed ~8 hours post-event. While existing models
provided a good overall fit and illuminated the divide between epikarst-to-conduit and matrix-
to-conduit flows, they were less effective in matching the long tails.
Accurate modeling of water movement in these complex subsurface landscapes is crucial, as
many regions rely on karst systems for drinking water (Stevanović, 2019). Here, a theoretical
and practical development of the CTRW framework is proposed as an approach to simulate the
intricate dynamics of water movement in karst environments.
**2 Conceptual and mathematical development**
The conceptual development of the CTRW framework to model water flow in karst systems is
founded on a proposed ansatz in which water flow is conceptualized as distinct "water parcels"
that travel along the available flow paths. Local volumes along the flow paths, e.g., caverns,
conduits, and voids, allow for the accumulation and release of water parcels, and define mobile
and immobile zones for water flow. The ansatz asserts that the accumulation and release of
water parcels in the various volumes in the karst system resemble the accumulation and release
of a chemical tracer over time in a porous medium. As shown in Fig. 1, a cavern acting as a
storage region for water parcels is analogous to tracer parcels accumulating in an immobile (or
less mobile) zone. For both cases, it should be noted that local accumulation of water parcels
or increase in concentration of a chemical will increase their respective fluxes in the immediate
local vicinity. Under similar hydraulic conditions both fluxes create distinctive long tails when
measured over a control plane at the system outlet, which is primarily a result of the structural
heterogeneity of the system.
Characterizing the flow of water through an infinitesimal control volume can be formulated in
terms of a balance equation that equates the net rate of fluid flow in the control volume to the
time rate of change of fluid mass storage within it:
$$-\frac{\partial(\rho q_x)}{\partial x} - \frac{\partial(\rho q_y)}{\partial y} - \frac{\partial(\rho q_z)}{\partial z} = \frac{\partial(\rho n)}{\partial t} \tag{1}$$

where $n$ is porosity, $\rho$ water density, and the three components of the specific discharge $q$ are
described as $q_x$, $q_y$ and $q_z$. This equation describes the mass balance in a fully saturated domain,





in which the void volume ($V_v$) is completely filled with water ($V_w=V_v$). The moisture content
($\theta = \frac{V_w}{V_{tot}}$) in these cases is equal to the porosity, and the degree of saturation ($\theta' = \frac{\theta}{n}$) is equal
to 1.
For partially saturated flow, the degree of saturation is less than 1 and the moisture content is
smaller than $n$ (as $V_w<V_v$). Adjusting the equation for partially saturated transient flow yields:
$$-\frac{\partial(\rho q_x)}{\partial x} - \frac{\partial(\rho q_y)}{\partial y} - \frac{\partial(\rho q_z)}{\partial z} = \frac{\partial(\rho\theta'n)}{\partial t} \quad . \tag{2}$$

Substituting $\theta' = \frac{\theta}{n}$:
$$-\frac{\partial(\rho q_x)}{\partial x} - \frac{\partial(\rho q_y)}{\partial y} - \frac{\partial(\rho q_z)}{\partial z} = \frac{\partial(\rho\theta)}{\partial t} \quad . \tag{3}$$

Deriving a description for a chemical tracer transport in a similar control volume is achieved
by a mass balance equation:
$$-\frac{\partial F_x}{\partial x} - \frac{\partial F_y}{\partial y} - \frac{\partial F_z}{\partial z} = n\frac{\partial C}{\partial t} \quad . \tag{4}$$

The chemical mass flux (in one direction) is defined by advection and diffusion terms:
$$F_i = q_i C - nD_i\frac{\partial C}{\partial i} \quad . \tag{5}$$

Substituting (5) into (4) yields
$$\frac{\partial}{\partial x}\left(nD_x\frac{\partial C}{\partial x}\right) - \frac{\partial}{\partial x}(q_x C) + \frac{\partial}{\partial y}\left(nD_y\frac{\partial C}{\partial y}\right) - \frac{\partial}{\partial y}(q_y C) + \frac{\partial}{\partial z}\left(nD_z\frac{\partial C}{\partial z}\right) - \frac{\partial}{\partial z}(q_z C) = n\frac{\partial C}{\partial t} \quad . \tag{6}$$

By drawing the analogy in the ansatz between the dynamics of water parcels and chemical
tracers, and noting the similar forms of Eqs. (3) and (4), the description of the mass balance of
water in a partially saturated domain is (at least) mathematically analogous to the description
of the mass balance of a chemical tracer in a saturated domain. This results in the intrinsic
connection of $C \Leftrightarrow \rho\theta$. In a 1D direction, the analogy of the mass flux can be thus defined:
$\rho q_x \equiv nD_x\frac{\partial C}{\partial x} - q_x C$. This connection incorporates hydrodynamic dispersion, which is
inherent in chemical transport resulting in observed long tails, into the description of the
partially saturated water parcels moving within the conceptual karst domain. Thus, the analogy
of chemical transport and water flow is expected to show long tailing in simple flow scenarios,
and was established even for pure diffusion (Cortis and Knudby, 2006).






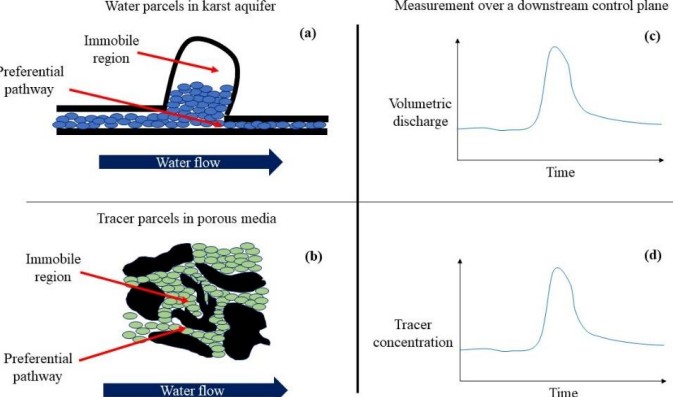


**Figure 1.** *Schematic illustration of (a) water parcels (blue ovals) in a karst aquifer; and (b) chemical tracer parcels (green ovals) in porous media (black grains) flowing through preferential pathways and accumulating in adjunct immobile regions. The resulting (schematic) measurements of the (c) temporal volumetric discharge and (d) tracer concentration that are measured at the spring outlet further downstream.*

Thus, transport equations – either advection-dispersion equations (ADE; Eq. (6)) for Fickian transport, or a CTRW formulation for non-Fickian transport (see Sect. 3.1) can be used, where the chemical tracer concentrations that these equations solve for $C(x,t)$ are conceptually identical to the relative concentration of water parcels. The concentration at a specific point is translated to a volume of water, so that a classical breakthrough curve $C(t)$ is reinterpreted as the (volumetric) amount of water per time reaching the domain outlet (or measurement plane).

## 3 Methods

### 3.1 CTRW-PT simulations

In this study, a particle tracking (PT) implementation of the CTRW framework was employed to devise a model capable of simulating spring discharge using the rainfall data as input. The CTRW-PT model, characterized by stochastically defined particle transitions, is a Lagrangian approach to solving the partial differential equations defined in the CTRW mathematical framework. The movement of the particles, representing water parcels as described in the ansatz (see Sect. 2), is described by equations that define the probability of particles to make




transitions in both space and time (Elhanati et al., 2023). For 1D cases, the transport is governed
by two probability density functions, $p(s)$ and $\psi(t)$, which define the particle movement in space
and time, respectively. An exponential from for $p(s)$ and a truncated power law (TPL) form for
$\psi(t)$ are used:

$$p(s) = \lambda_s^2 \exp(-\lambda_s s) \ , \tag{7}$$

$$\psi(t) = C \, \frac{\exp(-t/t_2)}{(1+t/t_1)^{1+\beta}} \ . \tag{8}$$

Here, $\lambda_s^2$ and $C$ serve as normalization factors for $p(s)$ and $\psi(t)$, respectively. The TPL is
governed by $\beta$, the power law exponent ($0 < \beta < 2$), which is a measure of the non-Fickian
nature of the transport, $t_1$, the characteristic transition time, and $t_2$, the cutoff time to initiate
transition to Fickian transport. The particle velocity, $v_\psi$, and the generalized dispersion, $D_\psi$,
are defined as the first and second spatial moments of the chemical species plume in the flow
direction (Berkowitz et al., 2006) For a 1D system:

$$v_\psi = \frac{\overline{s_x}}{\overline{t}} = \frac{\int_0^\infty p(s)s^2 ds}{\int_0^\infty \psi(t)t dt} \ , \tag{9}$$

$$D_\psi = \frac{1}{2}\frac{\overline{s_x^2}}{\overline{t}} = \frac{1}{2}\frac{\int_0^\infty p(s)s^3 ds}{\int_0^\infty \psi(t)t dt} \ , \tag{10}$$

where $\overline{s_x}$ $and$ $\overline{t}$ are the mean step size and time, respectively.
Inserting the probability density functions (Eqs. 7 and 8) into Eqs. 9 and 10, and defining $\tau_2 \equiv$
$t_2/t_1$ yields a mathematical relation among $v_\psi, D_\psi, \beta, \tau_2, t_1, t_2$ and $\lambda_s$ (see Nissan et al., 2017
for a full mathematical development). By treating the first four variables ($v_\psi, D_\psi, \beta, \tau_2$) as
fitting parameters, the other three ($t_1, t_2, \lambda_s$) are immediately determined, allowing
optimization of the CTRW-PT model to a specific flow scenario (see Table 1).
The intricate three-dimensional flow field of a karst system can be conceptualized in a model
that considers the relationships between storage and discharge. These kinds of models, known
as lumped models, have been extensively used in simulation of karst systems (Hartmann et al.,
2014). Herein, a similar approach is applied, i.e., conceptualizing the system as a series of
specific physical transitions. However, in the context of the CTRW-PT model, an equivalent
medium to the karst system is defined in the form of a one-dimensional domain. Water is
introduced into the domain along its entire extent and flows to the domain outlet.
The 1D conceptualization is facilitated by the well-defined spatial characteristics of the system,
namely the catchment area and spring outlet (Fig. 2a). The distance of each point on the surface
of the catchment to the spring outlet is calculated (Fig. 2b), which yields a distances frequency
histogram (Fig. 3). A normal distribution, fitted to the histogram using MATLAB, dictates how
new particles are introduced into the system along the 1D domain (physically unrealistic,
negative sampled values are set to 0). The actual underground flow path between each point
and the outlet spring is longer than the linear distance between the two points, as the water
must travel through the tortuous path through the existing conduits and fissures. The distances
are therefore multiplied by an empirical tortuosity factor ($L$), which serves as an optimization
parameter (see Table 1).

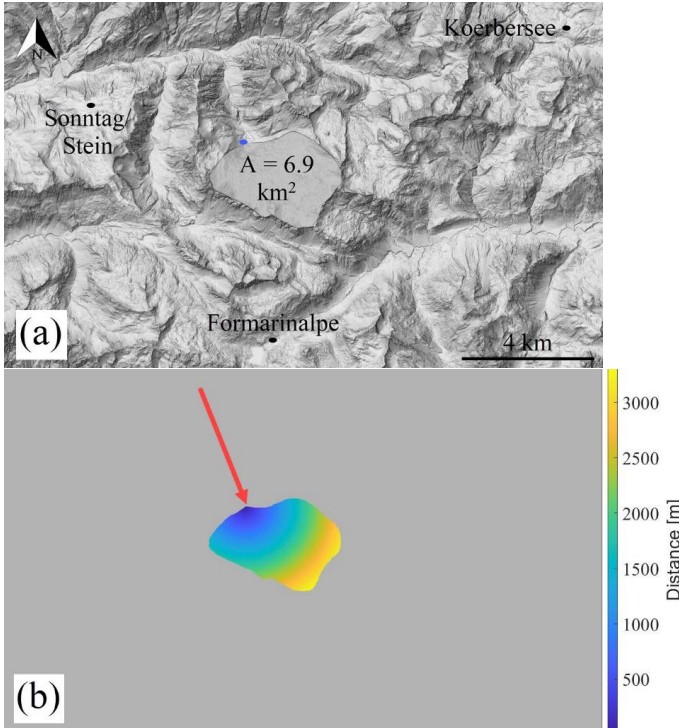

**Figure 2.** (a) *Map of the Disnergschroef study area. The three weather stations in which rainfall*
*was measured are marked with black dots, the measured spring outlet is marked with a blue*
*dot (basemap: Land Vorarlberg – data.vorarlberg.gv.at); (b) Distances from the catchment*
*area to the spring outlet. The distances are marked by a color scale. The spring outlet is marked*
*by a red arrow.*

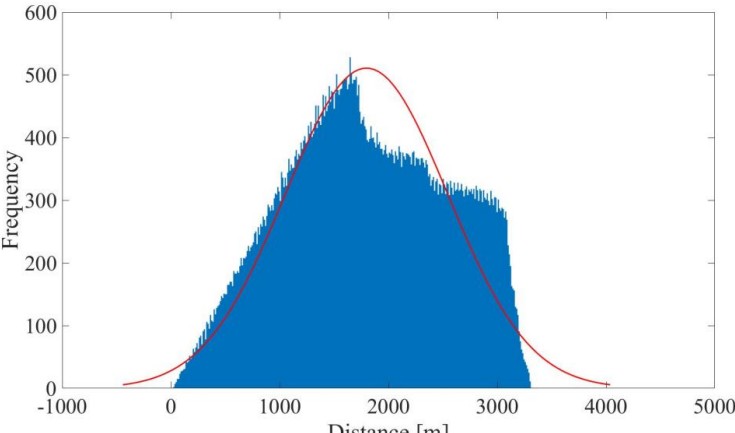


**Figure 3.** *Distribution of distances from catchment area to spring outlet. The red line represents a fitted normal distribution ($\mu=1.8\times10^3$; $\sigma=747$).*

Discharge at the spring is sampled every 15 minutes (L s$^{-1}$). The discharge minimum represents baseflow conditions. Raw rainfall data from three nearby weather stations (Fig. 2a) are measured in millimeters per 15 minutes. The data from the three stations are averaged, and the catchment area is used to convert the data into liters per second (Fig. 4). To achieve higher temporal simulation resolution, linear interpolation was used to resample the time series to match a smaller simulation time step (100 s).


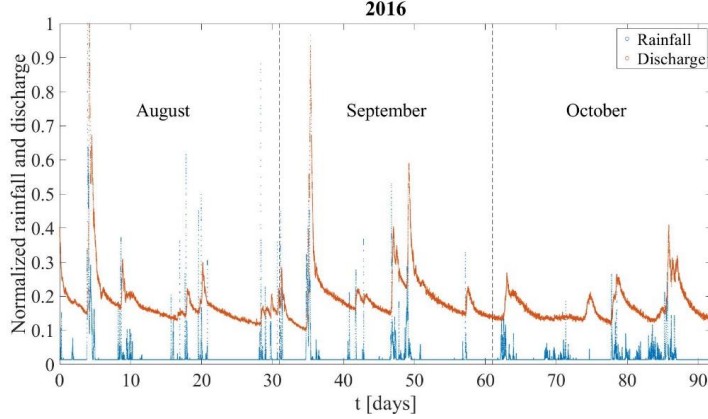




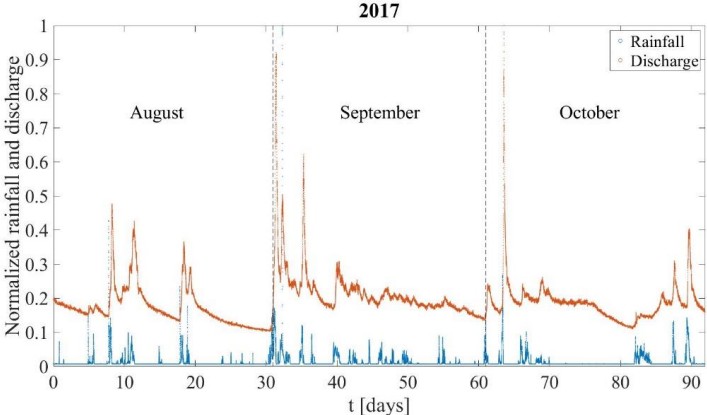


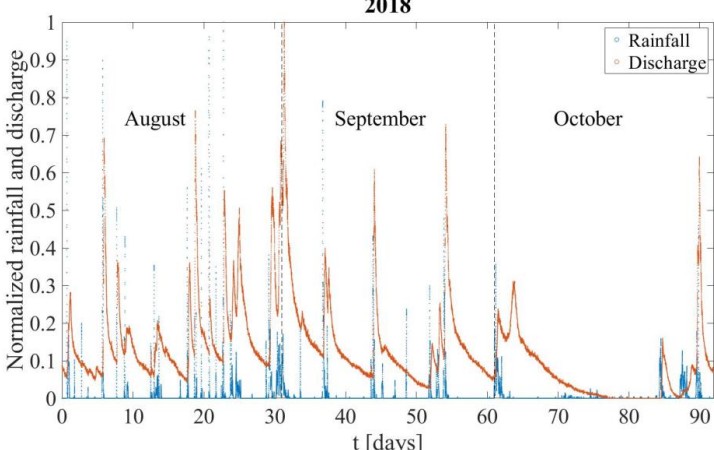

**Figure 4.** *Rainfall and discharge curves for the 2016, 2017 and 2018 datasets. The data are*
*normalized according to the maximum rainfall and discharge values, respectively, for each of*
*the three years.*

For an ideal system, in which all incoming rainwater is discharged through the spring outlet,
the ratio of total rainwater to total discharge is expected to be unity. However, considering the
uncertainty in the contributions of hydraulic parameters to the catchment water budget, e.g.,
flow to deeper parts of the aquifer and/or other springs, and evapotranspiration, the rainfall
function must be adjusted by a calculated observed recharge capacity to yield the recharge
function:
$$recharge(t) = rainfall(t) \times \frac{\sum discharge(t) - baseflow}{\sum rainfall(t)} \qquad (11)$$





where *rainfall*(*t*) and *discharge*(*t*) are the measured rainfall and discharge time series. The ratio
multiplying the rainfall function is defined here as the recharge capacity parameter. The
baseflow was subtracted from the total discharge for the recharge capacity calculation, to
account for the background discharge not related to the spring response to rainfall.
A common procedure in lumped karst models separates the flow into slow and fast components,
representing the diffusive flow in the matrix and smaller fissures and the rapid flow in the
conduits, respectively (Hartmann et al., 2014). The CTRW-PT, as opposed to lumped models,
does not utilize water flow reservoirs, and operates by tracking the motion of particles that
represent water parcels. Therefore, the model was adapted to implement a similar approach:
two different sets of CTRW parameters, which govern the probability density functions for
particle movement (see Eqs. 7-10), are defined to represent the two flow regimes. Each particle
in the simulation is defined as "slow" or "fast", and therefore obeys the corresponding set of
CTRW parameters (see Table 1). Newly introduced particles are divided between fast and slow
flow, according to a set ratio ($SF_r$), and they advance in space and time by their corresponding
set of CTRW parameters. Furthermore, each slow particle has a likelihood to transition into a
fast particle ($SF_l$) in each simulation iteration, by changing the set of CTRW parameters that
the particle obeys. The transition from slow to fast flow illustrates the flow of water from the
matrix/fissures to the conduits.
As depicted in Fig. 5, the likelihood of particle transition increases rapidly, with slow particles
consistently transitioning into fast particles. For a transition likelihood of 0.01% and a
simulation time step of 100 s, the likelihood for a single particle to make a transition surpasses
99% after 458 steps which amount to 45,800 seconds (~12.7 hours). In comparison, the data
and simulations presented in this study span a duration of 7,951,400 seconds (~92 days). These
two parameters, governing the division of water between fast and slow flow and the transition
of water from the matrix/fissures to the conduits, are pivotal in allowing the CTRW-PT model
to simulate karst data.

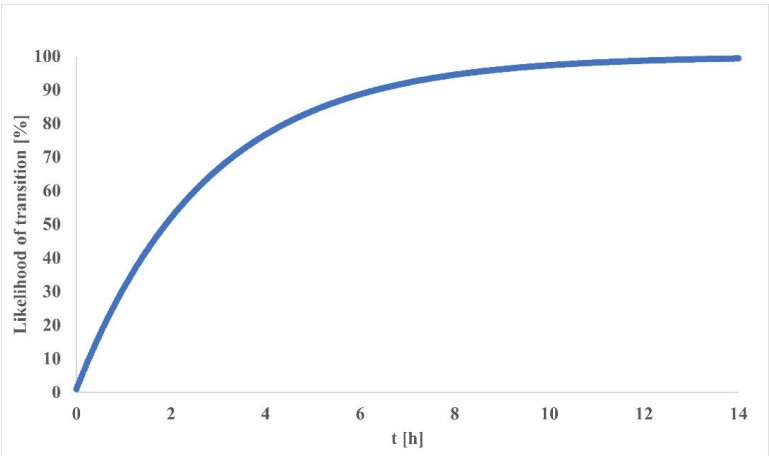

**Figure 5.** *Likelihood of particle transition from slow regime to fast regime (SF_l) as a function*

*time, representing a particle transition from slow matrix/fissure flow to fast conduit flow.*

### 3.2 Model optimization and comparison to field measurements

Each particle represents a volume of water. The volume per particle was calculated by dividing the total observed rainfall volume by the number of simulation particles. This enables a comparison between simulated and observed recharge by volume. Given the presence of numerous model parameters (refer to Table 1), optimization is achieved by minimizing the Root Mean Squared Error (RMSE) between observed and simulated discharge using different combinations of parameter values. The 2016 dataset was first utilized for parameter optimization, while the 2017 and 2018 datasets served as targets for validation, by considering them for prediction using the optimized parameters from the 2016 dataset.

The Nash-Sutcliffe efficiency (NSE) was calculated for the optimized simulations, as a measure of the goodness of fit. It is described as a normalized variant of the mean squared error:

$$NSE = 1 - \frac{\sum (x_s(t) - x_o(t))^2}{\sum (x_o(t) - \mu_o)^2} \tag{12}$$

where $x_s$ is the simulated discharge, $x_o$ is the observed discharge and $\mu_o$ is the observed mean. The NSE performance criterion is widely used in hydrological studies and does not induce counterbalancing errors (Cinkus et al., 2023a).



## 4 Results and Discussion

### 4.1 Optimized simulations of measured discharge

The optimized simulation for the 2016 dataset yields a fit (Fig. 6) that captures both the rapid response of the spring discharge to rainfall events and the protracted relaxation times characterized by the long tails evident after rainfall events. The optimized model parameters for the slow diffusive and fast conduit flow components are detailed in Table 1.

**Table 1.** *Optimized model parameters.*

| Parameter | Optimized value | Description |
|---|---|---|
| $v_\psi^f$ | 360 m h$^{-1}$ | Fast $v_\psi$ |
| $D_\psi^f$ | 36 m$^2$ h$^{-1}$ | Fast $D_\psi$ |
| $\beta^f$ | 1.7 | Fast $\beta$ |
| $\tau_2^f$ | $10^6$ | Fast $\tau_2$ |
| $v_\psi^s$ | 18 m h$^{-1}$ | Slow $v_\psi$ |
| $D_\psi^s$ | 3.6 x 10$^8$ m$^2$ h$^{-1}$ | Slow $D_\psi$ |
| $\beta^s$ | 1.2 | Slow $\beta$ |
| $\tau_2^s$ | $10^8$ | Slow $\tau_2$ |
| $L$ | 1.6 | Tortuosity |
| $SF_r$ | 0.95 | Slow to fast particle ratio |
| $SF_l$ | 0.01 % | Slow to fast particle transition likelihood |

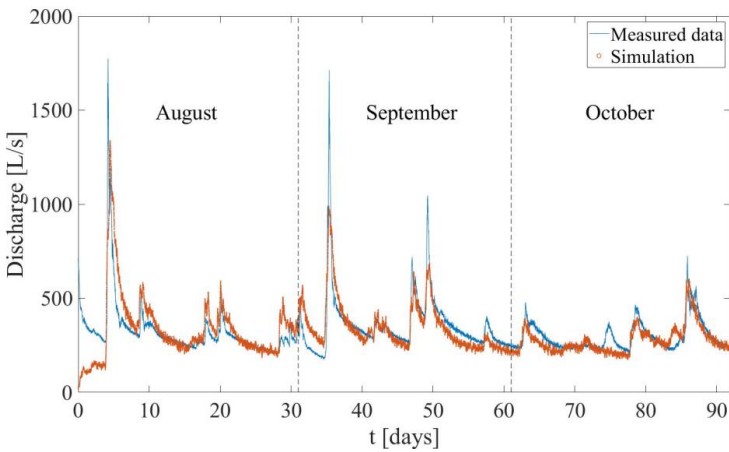

**Figure 6.** *Measured and simulated spring discharge for the 2016 dataset (NSE=0.5).*





The differences between the fast and slow flow components, as illustrated by the respective
optimized CTRW parameters, elucidate the contribution of each flow component to the
volumetric discharge. The fast flow velocity parameter ($v_{\psi}^{f} = 360\ m\ h^{-1}$) is much larger than
the slow flow velocity parameter ($v_{\psi}^{s} = 18\ m\ h^{-1}$), and shows how incoming rain can rapidly
flow to the spring outlet, when travelling through the large conduits. The slow diffusive flow,
however, has a much longer travel time than the fast flow. Another clear difference between
the two components which is evident from the optimized values is the degree of anomalous
transport. The fast flow $\beta$ (1.7) and $\tau_{2}$ ($10^{6}$) parameters lead to a more symmetrical contribution
to the resulting discharge around the peak following the recharge event, compared to the slow
flow parameters ($\beta$=1.2, $\tau_{2}$=$10^{8}$), which create a long tail after the discharge peak. The slow
flow is also much more dispersive ($D_{\psi}^{s} = 3.6\ x\ 10^{8}\ m^{2}\ h^{-1}$) compared to the fast flow ($D_{\psi}^{f} =$
$36\ m^{2}\ h^{-1}$), which contributes further to the long discharge tails. The optimized parameters
show a strong prominence of the slow flow over the fast flow: 95% of newly introduced
particles are introduced as slow particles ($SF_{r}$), with a 0.01% likelihood for a slow particle to
transition at each iteration to a fast regime ($SF_{l}$).
The fit obtained for the 2016 dataset modeling is satisfactory considering the inherent
uncertainty associated with the input data. The three weather stations used to measure the
precipitation are not located inside the catchment area, and different precipitation data were
measured at each station, which can be seen by examining the cross-correlation coefficients
between the 2016 discharge and rainfall data: 0.20, 0.22 and 0.15 for stations Koerbersee,
Formarinalpe, Sonntag/Stein, respectively (Fig. 2a). While an average of the three stations
provides an acceptable estimate of the recharge over the given time period, the variability of
local rain events is overlooked, which may be common in the high mountainous topography.
This is especially true in extreme rain events, in which variations of onset, duration, and total
discharge of an event can induce different responses of the modeled discharge.
The same set of CTRW parameters optimized for the 2016 data – without further adjustment –
was employed to interpret the 2017 and 2018 datasets (Fig. 7). Both datasets show that the
simulated discharge after rainfall events predicts the onset, length and volume of the measured
discharge. This is especially true for the many discharge peaks exhibited by the 2018 data.

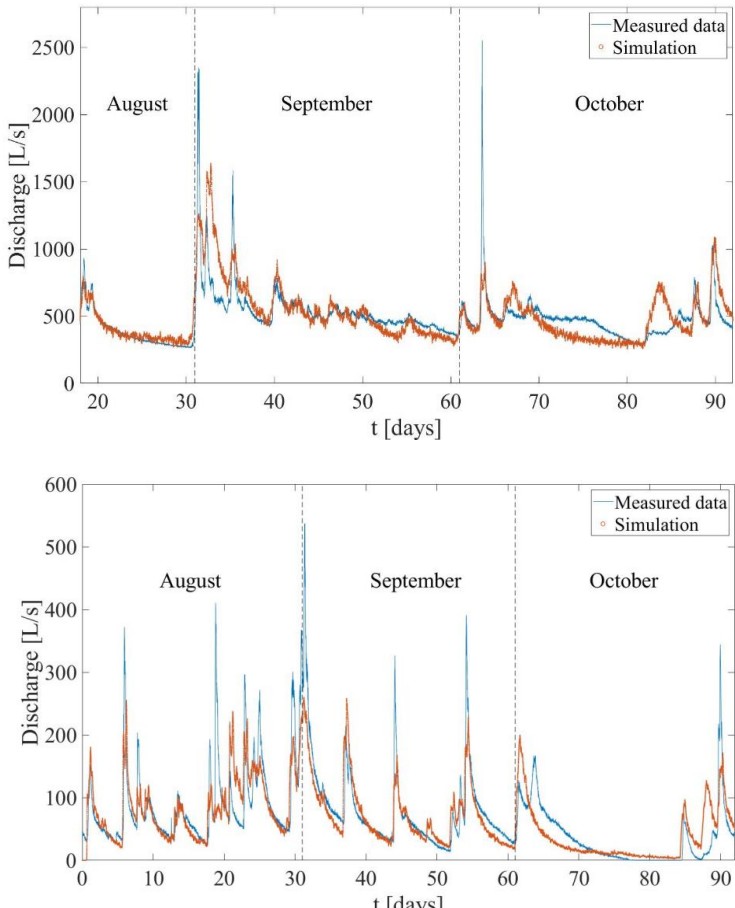


**Figure 7.** *Measured and simulated spring discharge for the 2017 (top; NSE=0.33) and 2018*


*(bottom; NSE=0.63) datasets. Note that due to the large differences in maximum discharge*


*between the three years, the vertical scales in Fig. 6 and in Fig. 7 are adjusted accordingly.*



The recharge capacity parameter (applied to calculate the recharge function from the measured
rainwater; see Eq. 11) was calculated as 0.43 and 0.45 for the 2016 and 2017 datasets,
respectively. These values suggest that ~40% of the incoming rainfall reaches the outlet spring,
with the remaining water reaching deeper parts of the aquifer that are less mobile. The drier
2018 dataset, however, displayed a much lower value of 0.19. The variability of the recharge
capacity parameter in different time periods, as a function of the rainfall pattern and amount,



highlights the importance of this parameter to the correct prediction of the system discharge
response to rainfall.
**4.2 Prominence of the slow flow component in the Disnergschroef system**
The prominence of the slow component in this karst system is evident from both the high $SF_r$
and low $SF_l$. The consistency of this finding, across the three datasets (Fig. 6 and 7), agrees
with the analysis by Frank et al. (2021) of the recharge/discharge relationship. They observed
that while the flow from epikarst to conduit and matrix is highly variable and rainfall-
dependent, the matrix to conduit flow remains relatively constant up to a threshold. The
coupling of the two flow processes produces a distinctive discharge pattern characterized by a
sharp rapid peak after a rainfall event, followed by a long tail during recession and a return to
baseflow. The current analysis is similar and further emphasizes that the volumetric
contribution of the slow flow is substantial, particularly influencing the extended tails. In
contrast, the fast flow plays a more straightforward role, contributing predominantly to
discharge peaks by quickly expelling introduced rainwater from the system.
Given the importance of karst systems for human consumption, monitoring and prediction of
system discharge is especially important during high and low flow scenarios. These extreme
events can have consequences on water quality, including over-consumption during dry periods
and increases in turbidity and bacterial activity in high flow conditions (Pronk et al., 2006).
The frequency of both dry periods and heavy rainfall events has been shown to rise due to
climate change (Stoll et al., 2011), and this may well increase in the near future. In this context,
the high peaks and long tails associated with these flow conditions have proven to be the most
difficult to correctly predict, across different karst modeling approaches (Jeannin et al., 2021).
The results presented of the CTRW modeling exhibits the long tails associated with low water
flow. The 2018 dataset, in particular, which represents a dry summer compared to the other two
datasets, exemplifies the robustness of the model in predicting low flow conditions.
**4.3 The contribution of the slow and fast flow components to simulated discharge**
The results for all three datasets do not show agreement between the maximum simulated and
observed discharge values that are found immediately after high recharge events. The better fit
of the long tails compared to the high peaks is evident in the improvement of the NSE values
presented above (0.50, 0.33, 0.63 for 2016, 2017 and 2018, respectively), when calculated for
the data without the prominent peaks (0.75, 0.60, 0.65 for 2016, 2017 and 2018, respectively).
The dry 2018 dataset is the least affected from removing the peak for the NSE calculation





because the peaks are low relatively to the 2016 and 2017 datasets. The fast response of
discharge to the incoming rain in karst systems after high recharge events has been described
in previous studies as a piston effect (Aquilina et al., 2006; Hartmann et al., 2014). Incoming
rain creates a rise in discharge before the rainwater reaches the outlet, as the increase in
hydraulic head pushes out water that was retained in the system before the rain. This effect was
shown specifically in the Disnergschroef system by Frank et el. (2021) which measured a 2.5-
hour difference between the first response of spring discharge to a rainfall event, to the arrival
of the rainwater to the outlet. The model herein does not take this effect into account, which
creates the negative bias in modeling the high peaks. While outside the scope of this study, this
can be addressed in the future by adding a third flow component, or by altering the CTRW
parameters of the particles present in the system prior to the rainfall event to represent the
increase in flow velocity.
To further examine the effect of both the slow and fast flow components on the simulated
discharge, simulations that examine the $SF_l$ and $SF_r$ parameters across a wider range were
conducted (Fig. 8). Simulations that contained only fast or slow particles (Fig. 8a), clearly show
that fast flow discharge responds very quickly to rainfall and produces no observable tails. In
contrast, the slow flow produces very long tails. It is noteworthy that the first response of the
slow flow is similar to the fast flow, as particles that are introduced to the system close to the
outlet have a very short length to travel to reach the outlet. Mixing of both flow regimes, either
by directly splitting the particles between the two regimes as they are introduced (Fig. 8b) or
by changing the transition likelihood (Fig. 8c) produces an intermediate response: as more of
the flow is slow, longer tails are found but the peaks are smaller.

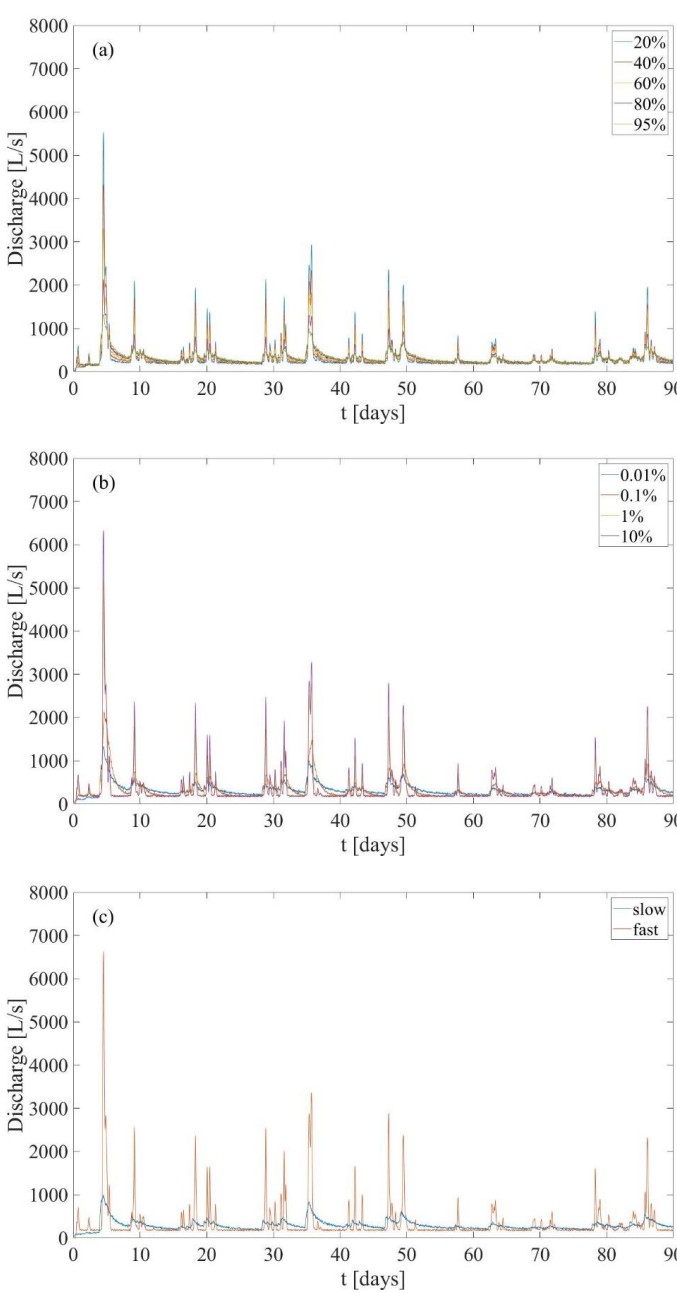


**Figure 8.** *Simulation sensitivity to slow and fast particle contribution, based on the 2016*
*rainfall data. Simulations containing only one kind of particle (a), Slow/Fast ratio (b) and*
*transition likelihood from slow to fast (c) demonstrate the importance of the slow flow for the*
*observed long tails in the discharge data.*



## 5 Conclusions

An analogy between partially saturated water flow in karst aquifers and anomalous chemical transport is established, allowing for the adaption of the CTRW-PT model to water flow in general, and for karst discharge response to rainfall specifically. The model was calibrated on one summer season of measurements of spring outlet discharge response to incoming rain; it was then used to predict the long tails observed in discharge measurements following rainfall events in two subsequent summer seasons.

The investigation of the Disnergschroef karst system has showed that slow diffusive flow is a predominant contributor to the volumetric discharge response to recharge events, in comparison to fast conduit flow. This finding highlights the nuanced interplay between fast and slow flow components in karst systems, and how they both evolve over time and as a function of the recharge intensity.

The theoretical and practical advancements presented here offer a potentially robust tool to further assess long-tailed rainfall-discharge responses in karst systems and other complex, catchment-scale systems.



**Data availability**

The data on which this article is based are available online on Zenodo: https://zenodo.org/doi/10.5281/zenodo.10635639 (Elhanati and Berkowitz, 2024).

**Author contribution**

DE, NG and BB formulated the ideas which originated the project and defined the goals and aims of the study. DE developed and implemented the methodology and carried out the data analysis. DE and BB drafted the initial manuscript. All authors took part in reviewing and editing the final manuscript.

**Competing interests**

BB is a member of the editorial board of the journal.

**Acknowledgments**

We thank Yael Arieli for helpful insights during the practical adaption of the CTRW-PT model in the course of this study, and Simon Frank for sharing the original data sets and initial background on the field measurements. DE and BB gratefully acknowledge the support of the Weizmann Institute for Environmental Sustainability and the Israel Science Foundation (grant No. 1008/20), respectively. BB holds the Sam Zuckerberg Professorial Chair in Hydrology. NG thanks the Water Management Department of the Vorarlberg State Administration for providing rainfall data.

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
