# Peer review of "Karst aquifer discharge response to rainfall interpreted as anomalous transport"

_Hydrology and Earth System Sciences, 2024_

## Referee Comment (RC1)

**Review: Karst aquifer discharge response to rainfall interpreted as anomalous transport**

The authors simulate flow dynamics in a karst system via a CTRW particle tracking approach. The approach is based on the idea of representing a flow in (partially) saturated domain analogous to transport dynamics involving accumulation and release of tracer substances. The authors demonstrate that the model is able to recover the combined effects of both the slow and rapid flow components by using suitable parameterizations for the probability density functions governing particle movement in space and time of each domain. Furthermore the authors demonstrate limitations of the approach which is attributed to piston-type flow processes that could potentially be modeled via a third component.

The paper is concise (partially it could be slightly extended) and I could follow the main points of the authors. The graphics are mostly of high quality. In my manuscript figures showing discharge (e.g. 4, 6...) are partially blurry, though this may be a problem of my pdf viewer or the manuscript draft quality.

In the following I have a few content related remarks. I have not found typographic errors.

**Content:**

1. (line 209) Flow paths are assumed to be linear elements connecting each point in the catchment with the spring. With a tortuosity factor unknown features of the flow geometry are taken into account. While this approach is efficient it potentially neglects various aspects of the internal system geometry. Here or in the discussion it would be helpful to add some further information about the catchment (if available). What is the thickness of the vadose zone and roughly what volume of the system is considered phreatic? While the chosen flow paths approach may be more realistic for systems which negligible vadose zone and/or thin phreatic zone in other cases a more realistic flow paths distribution may be chosen. Małoszewski and Zuber (1996, 2001) in their works have for example

often assume various flow distribution patterns (though mostly assuming phreatic settings).

2. (line 248) Recharge is estimated with a rather simple approach neglecting more complex processes in the soil water balance or runoff dynamics. How is baseflow computed? The authors should briefly discuss how the simplification might influence the results. Certain effects of this process spectrum might for example affect the peak recharge and hence discharge dynamics (e.g. more runoff might decrease the input during strong precipitation events). As the peak discharge components are difficult to fit without a third component according to the authors this may already be one potential issue here.

3. (line 266) My understanding is that particles are able to transition from the fast into the slow domain. While this is conceptually often the case, in the bulk aquifer system the opposite case can occur in both the phreatic and especially vadose zone. Karst conduits connected via rapid infiltration features to the surface (dolines, sinks) may transmit water during recharge events into the matrix when a gradient inversion occurs. In addition within the vadose zone typically the matrix receives water from adjacent rapid flow elements (e.g. fractures) due to the differences in capillary pressure, (though this effect may not be present under non-equilibrium conditions as preferential flowpaths may partially overcome the effect of capillary suction). This limitation in the approach should be briefly mentioned in the discussion section.

4. (line 374) "without the prominent peaks" What is exactly meant by this? Are you removing exactly one datapoint (15min intervals)?

5. (line 386) Why would this need a third component? Is it impossible to fit this with the available parameters or possibly a different form instead of the TPL? Many karst springs have been successfully modeled with dual-domain approaches (spatially distributed, lumped parameter approaches). I wonder wether this is a limitation of the approach or caused by the distribution of heterogeneities (fractures, conduits) specific to this spring system.

6. Conclusion chapter: Here I feel it would be good to slightly extend the scope and relate the results to other modeling approaches. Why should one employ the demonstrated approach? What are the benefits in terms of process representation and computationally efficiency?

---

## Referee Comment (RC2)

**Title:**

Karst aquifer discharge response to rainfall interpreted as anomalous transport

**Authors:**

Dan Elhanati, Nadine Goeppert, Brian Berkowitz

**General remarks:**

The proposed methodology have a significant interest among the present challenges in karst hydrology for flow and transport modeling. The manuscript is of interest for the scientific community. Nonetheless, I found some important methodological aspect that, in my point of view, should be addressed before potential publication.

**Specific comments to the paper:**

Line 62: the role of the pore connection should be also mentioned, porosity is an important factor in flow and transport as well as the continuity between pores. This notion should appears in the introduction, as well as tortuosity, which then used in the proposed methodology.

Line 104: "well-defined spatial catchment" sounds unclear, are the boundaries well defined with the topography and/or geological setting?

Line 117 "ansatz" is a German word, this appears several time across the manuscript.

Line 117: "water parcels" the notation sounds unclear is it like "water bodies" between flow zone and dead zone or is it a quantity of water that should be considered as an analogy with an amount of particle ? This point can be clarified in the paragraph.

Line 123: "water parcels" and "tracer parcels" sounds unclear, a more detailed explanation might help the reader to understand how the analogy is done in the manuscript.

Fig 1. (c) and (d) does the volumetric discharge considers a constant concentration and does the concentration variation assumes a constant discharge ? In my point of view, discharge and concentration cannot be separated so the two might appears on the graph, then the final time series that the author are using will appears more clear to the reader. At some point, the present version bring some confusion with the widely used tracer BTC.

Line 172: "C(t) is reinterpreted as translated to a water volume" please clarify the underlying hypothesis to move from concentration curve to fluxes curve.

Fig 3. I don't think we can consider the fit as satisfactorily. The data do not exhibit a normal distribution shape, also the fitted distribution provides negative distances. The fitted distribution should then at least be a truncated distribution to avoid negative values.

Fig 5. How is the curvature of the likelihood distribution fixed? Is it a working hypothesis or is it derived from somewhere else?

Line 283: 'optimization is achieved' How is the optimization performed? Which methodology/algorithm? I would recommend also to write "model parameter estimation" rather than "optimization".

Line 293: Why using NSE values that tends to favorize large values while the focus seems to be on tailing? Other performance criteria for model evaluation would be more suitable, or another option would be to compute NSE on a variable transformation such as 1/Q or squared_root(Q)

Table 1: Please add the min and max of the investigated parameter space for the model parameter estimation. Also, please give more information about how the optimized value is estimated (see one of the previous comment) + the estimated tortuosity appears pretty high compared with the literature (e.g. reference below) could you discuss a little on that ? Is it a realistic value for the catchment?

Jouves, J., Viseur, S., Arfib, B., Baudement, C., Camus, H., Collon, P., Guglielmi, Y., 2017. Speleogenesis, geometry, and topology of caves: A quantitative study of 3D karst conduits. Geomorphology 298, 86–106. https://doi.org/10.1016/j.geomorph.2017.09.019

Collon, P., Bernasconi, D., Vuilleumier, C., Renard, P., 2017. Statistical metrics for the characterization of karst network geometry and topology. Geomorphology 283, 122–142. https://doi.org/10.1016/j.geomorph.2017.01.034

Line 324: Is it correlation coefficient or correlation pic derived from cross correlation function analysis? What is the time lag considered for the correlation coefficient or what is the lag response obtained based on precipitation-discharge cross correlation function?

Line 328: This sentence is not useful. I recommend to delete.

Line 374: As previously mentioned in my comments, other performance criteria would be more suitable to evaluate the predictive performance of the model regarding high flow and low flow periods respectively. Computing NSE by removing the high value is not a suitable justification for the model improvement on low flow period. Indeed, by skipping the high discharge value to compute the NSE you are changing the benchmark of NSE (the mean of observed time series) so comparison of NSE is not straightforward in that case.

Line 377: Are there some evidence of piston effect with temperature and/or conductivity?

Fig 8. Is not very informative as "sensitivity analysis". Including a sensitivity analysis on the model parameter regarding the model performance would be much more informative. Among the model parameters which one are the more sensitive in the model?

---

## Author Response (AR1)

25 July 2024

To:

Editor— *Hydrology and Earth System Sciences*

Dear Editor,

RE: Manuscript HESS-2024-46:

Re-submission of the manuscript entitled "Karst aquifer discharge response to rainfall interpreted as anomalous transport" by D. Elhanati et al.

Dear Editor,

We thank you and the reviewers for the thorough reviews and very constructive comments. We have now revised the manuscript considering the reviewers' comments and suggestions.

Please find below our point-by-point responses, including details of how we addressed each of the reviewer comments in the revised manuscript.

We hope that you will find the revised version suitable for publication in Hydrology and Earth System Sciences.

Yours sincerely,

Dan Elhanati, corresponding author on behalf of all authors.

**Response to editor:**

1. The manuscript has been reviewed by two experts. Both reviewers find the submitted work to be of interest to HESS readers, provided moderate revisions are made. The authors have addressed all the major points raised during the review process in their rebuttal. In addition to the reviewers' comments, I encourage the authors to consider the following points:

Reply: We thank the editor for the constructive comments. See below our replies to the specific comments.

2. Analogy between equations (3) (flow in partially saturated media) and (2) (transport in fully saturated media): While this analogy is clear when the water density is constant, it becomes less straightforward if the density varies in time and/or space. Since this is not crucial for the manuscript, I suggest rewriting equation (2) assuming constant density. Note that equation (4) is written for constant porosity and can be simplified, making the analogy between equations (3) and (4) straightforward.

Reply: We believe the analogy between Eqs. (3) and (4) [not (2) as written above] is clear as written, allowing at least in principle for variable density. One can certainly assume incompressibility, but we prefer to retain the density term as written. The main reason for this is so that the analogy/ansatz $C \Leftrightarrow \rho\theta$ is consistent with both terms having units of mass per volume; Eqs. (3) and (4) compare the terms $\frac{\partial(\rho\theta)}{\partial t}$ and $\frac{\partial C}{\partial t}$. In light of the suggestions, we have modified the wording in the revised manuscript in three locations, to read:

"*Adjusting the equation for partially saturated transient flow yields (allowing for water compressibility, to retain generality):*" immediately preceding Eq. (2).

"*(Note that the appearance of the porosity variable, n, in the terms of Eqs. (4)-(6) is easily rearranged, and that these equations can be simplified if n is assumed constant.)*" immediately following Eq. (6).

and

"*This results in the intrinsic connection of $C \Leftrightarrow \rho\theta$, both with units of mass per volume.*" in the discussion of the ansatz following Eq. (6).

3. Lines 212 - 214. It is unclear why you need to fit a normal distribution to your data, which are clearly not normal. In your response to one reviewer, you state that "preliminary results were similar for different skewed distributions." Can you use the data directly without fitting a model?

Reply:

The calculated distances histogram is dependent on initial image resolution and chosen bin size. The main motivation of fitting a distribution is to avoid the dependence of the model on these two properties. Moreover, using the histogram directly would result in discrete particle entry locations, which is not realistic (as compared to a distribution over the continuum). A distribution function must be defined in order to distribute the particles along the non-discrete domain. We agree that after establishing the necessity of a distribution function, one must yet choose the correct distribution: normal or otherwise. As the model results proved to be insensitive to the choice of distribution, we chose to use a normal distribution for simplicity. We have revised the text to clarify this point (see also comment #8, reviewer 2): "*The 1D conceptualization is facilitated by the well-defined spatial characteristics of the system, namely the catchment area and spring outlet (Fig. 2a). The distance of each point on the surface of the catchment to the spring outlet is calculated (Fig. 2b), which yields a frequency histogram of distances (Fig. 3). The histogram shape is dependent upon the initial image resolution and the chosen bin size and yields discrete distances. To sample continuum particle entry locations without dependence on bin size, a distance distribution, fitted to the histogram using MATLAB, dictates how new particles are introduced into the system along the 1D domain (physically unrealistic, negative sampled values are set to 0). A normal distribution was chosen as a simplified representation of the distance distribution; preliminary simulation results were similar for different skewed distributions.*".

4. Table 1. I agree with one reviewer about the need to include more details (perhaps in a supplementary material) about how the optimization procedure was performed.

I would also recommend including the uncertainty associated with the optimized values. This additional information is relevant to what is noted in lines 212-214.

Reply: We have revised the text to include more details about the optimization procedure and included the minimum and maximum values of the investigated parameter space in Table 1, as noted by reviewer #2 (see comments #8 and #10 for the revisions).

5. Figure 8. The content of this figure is not clear, as also noted by a reviewer.

Reply: In the revised text, we have edited the figure caption to clarify the parameters for which the sensitivity analysis is relevant ($SF_r$, $SF_l$, and using only one type of particles). See also comment #17 (reviewer 2) for the revised caption: "***Figure 8. Simulation sensitivity to slow and fast particle contributions, based on the 2016 rainfall data. Simulations that compare different $SF_r$ values (a), different $SF_l$ values (b) and only fast/slow particles (c), demonstrate the importance of the slow flow for the observed long tails in the discharge data.***".

6. Line 130 "a balance equation". Please, specify that it is a mass balance equation.

Reply: Done, the text was revised accordingly: *"Characterizing the flow of water through an infinitesimal control volume can be formulated in terms of a mass balance equation that equates the net rate of fluid flow in the control volume to the time rate of change of fluid mass storage within it".*

7. Line 133 "q" is a vector and should be in bold

Reply: Done.

8. Line 143 Please rewrite this sentence, which introduce eq (4). Explicitly state here that Eq. (4) is for transport in fully saturated media (currently this information is only provided at line 153).

Reply: The sentence was rewritten to reflect that the chemical transport is in a fully saturated media: *"Deriving a description for the transport of a chemical tracer in a fully saturated porous medium within a similar control volume is achieved by a mass balance equation:".*

**Response to reviewer #1:**

**General comments:**

1. The authors simulate flow dynamics in a karst system via a CTRW particle tracking approach. The approach is based on the idea of representing a flow in (partially) saturated domain analogous to transport dynamics involving accumulation and release of tracer substances. The authors demonstrate that the model is able to recover the combined effects of both the slow and rapid flow components by using suitable parameterizations for the probability density functions governing particle movement in space and time of each domain. Furthermore the authors demonstrate limitations of the approach which is attributed to piston-type flow processes that could potentially be modeled via a third component.

Reply: We thank the reviewer for the positive appraisal of the manuscript.

2. The paper is concise (partially it could be slightly extended) and I could follow the main points of the authors. The graphics are mostly of high quality. In my manuscript figures showing discharge (e.g. 4, 6...) are partially blurry, though this may be a problem of my pdf viewer or the manuscript draft quality.

Reply: In the revised submission, higher quality figures will be submitted.

3. In the following I have a few content related remarks. I have not found typographic errors.

Reply: See below our comments for the specific content-related remarks.

**Specific comments:**

1. (line 209) Flow paths are assumed to be linear elements connecting each point in the catchment with the spring. With a tortuosity factor unknown features of the flow geometry are taken into account. While this approach is efficient it potentially neglects various aspects of the internal system geometry. Here or in the discussion

it would be helpful to add some further information about the catchment (if available). What is the thickness of the vadose zone and roughly what volume of the system is considered phreatic? While the chosen flow paths approach may be more realistic for systems which negligible vadose zone and/or thin phreatic zone in other cases a more realistic flow paths distribution may be chosen. Ma loszewski and Zuber (1996, 2001) in their works have for example often assume various flow distribution patterns (though mostly assuming phreatic settings).

Reply: We agree that the internal system geometry is very complex, and this is true for the fractioning of the aquifer to vadose and phreatic zones as the comment states, but also for the structural characteristics of both zones. We argue that both of these complexities are represented in our simulations by the choice of model parameters. However, we do agree that in presenting a single case study for testing our model we should further expand about the nature of the system. In the revised text, we added a more detailed hydrogeological description of the system to address this: *"The surface of the karst system is composed mainly of bare limestone with very limited soil coverage, resulting in negligible surface runoff. The plateau is characterized by dolines and depressions, further facilitating the direct flow of water into the subsurface. The vadose zone is estimated to be several hundred meters thick (Frank et al., 2021)."*. We also revise the discussion to point out that the thickness of the zones might play an important role in the calibration of the model: *"The $SF_l$ and $SF_r$ are thus important parameters as they allow application of the CTRW-PT model to different karst systems. The Disnergschroef system, presented here as a case study, is characterized by a thick vadose zone and negligible surface runoff. Different karst systems are likely to show different $SF_l$ and $SF_r$ parameters."*.

2. (line 248) Recharge is estimated with a rather simple approach neglecting more complex processes in the soil water balance or runoff dynamics. How is baseflow computed? The authors should briefly discuss how the simplification might influence the results. Certain effects of this process spectrum might for example affect the peak recharge and hence discharge dynamics (e.g. more runoff might decrease the input during strong precipitation events). As the peak discharge

components are difficult to fit without a third component according to the authors this may already be one potential issue here.

Reply: We agree that the estimation of recharge from rainfall is rather simple. It is estimated using a bulk ratio factor which considers the total discharge, rainfall and the baseflow (which is calculated from the minimum of the observed discharge). In the revised text, we expanded the methods section accordingly to clarify the following points:

a.  The baseflow is the discharge minimum: *"The minimum measured discharge represents the baseflow discharge"*.

b.  The rainfall to recharge ratio may be dependent upon time, rain intensity and spatial characteristics. The sensitivity of the three aforementioned quantities should be taken into account in future studies especially when they display high variability: *"While a constant recharge capacity factor is employed in this study, due to the negligible surface runoff, it is important to note that the rainfall-to-recharge ratio may be influenced by temporal variations, rainfall intensity, and spatial characteristics. Future research should consider the sensitivity of these variables for the specific scenarios considered. In cases where there is significant variability among them, other temporal and/or spatial ratios may be applied."*.

3.  (line 266) My understanding is that particles are able to transition from the fast into the slow domain. While this is conceptually often the case, in the bulk aquifer system the opposite case can occur in both the phreatic and especially vadose zone. Karst conduits connected via rapid infiltration features to the surface (dolines, sinks) may transmit water during recharge events into the matrix when a gradient inversion occurs. In addition within the vadose zone typically the matrix receives water from adjacent rapid flow elements (e.g. fractures) due to the differences in capillary pressure, (though this effect may not be present under nonequilibrium conditions as preferential flowpaths may partially overcome the effect of capillary suction). This limitation in the approach should be briefly mentioned in the discussion section.

Reply: We agree that both slow to fast and fast to slow transitions may occur. However, CTRW results are essentially an ensemble average, and represent processes as statistical outcomes. In the specific context, the chance of slow to fast transition represents the net total transition. If particles can also transition back from fast to slow, then this can be represented by simply lowering the chance of transition from slow to fast. We revised the text accordingly to address this point: *"While transition of fast to slow flow is also possible in karst aquifers, i.e., when the pressure gradient allows water from the conduits to enter the matrix, the slow to fast transition is more prominent for this site. Thus, the likelihood of transition represents the net transition from slow to fast flow. When more particles transition back from fast to slow flow, the transition likelihood is lower. In this context, it is important to note that the CTRW-PT is a stochastic approach, in which the system parameters are represented by statistical properties. The results of CTRW-PT simulations are, therefore, representative of an ensemble average of many realizations."*.

4. (line 374) "without the prominent peaks" What is exactly meant by this? Are you removing exactly one datapoint (15min intervals)?

Reply: In light of this comment and other comments made by reviewer #2, we have re-examined the peak removal, and decided to omit it from the manuscript.

5. (line 386) Why would this need a third component? Is it impossible to fit this with the available parameters or possibly a different form instead of the TPL? Many karst springs have been successfully modeled with dual-domain approaches (spatially distributed, lumped parameter approaches). I wonder wether this is a limitation of the approach or caused by the distribution of heterogeneities (fractures, conduits) specific to this spring system.

Reply: We agree that the option of adding a third flow component is not sufficiently clear in the text. The application of the CTRW modeling framework for karst flow is a new approach and this is the first implementation. Specifically, we applied it for this karst system as dual flow approaches did not adequately fit the observational data, essentially

because of the distribution of heterogeneities. The main purpose of the simulation was to demonstrate the ability to fit the extremely long tails which can be observed in many karst systems. In doing so, we highlight that the peaks are less in agreement, and therefore offer possible ways to deal with peak data in future studies. A third flow component is one option (as discussed in the text for celerity), but we do not think this is the only option or the most appropriate one. We therefore point out that this is a possibility that should be considered outside of the scope of this paper. We revised the text to better convey this message: *"The model herein does not explicitly take this effect into account, which creates the negative bias in modeling the high peaks. While outside the scope of this study, this feature might be addressed in the future by adding a third flow component, or by further refining the CTRW parameters of the particles present in the system prior to the rainfall event to represent the increase in flow velocity."*.

6. Conclusion chapter: Here I feel it would be good to slightly extend the scope and relate the results to other modeling approaches. Why should one employ the demonstrated approach? What are the benefits in terms of process representation and computationally efficiency?

Reply: We agree that the conclusions can be expanded to better convey the possible use of our model. The main purpose of the application of the CTRW model for this case (see reply #5 above as well), was to address the long tails in karst discharge data. We revised the text to convey this: *"The application of the CTRW-PT model for the Disnergschroef system, specifically, has shown that it is particularly advantageous in predicting the long tails observed in discharge data, compared to other modeling approaches."*.

**Response to reviewer #2:**

**General remarks:**

1. The proposed methodology have a significant interest among the present challenges in karst hydrology for flow and transport modeling. The manuscript is of interest for the scientific community. Nonetheless, I found some important methodological aspect that, in my point of view, should be addressed before potential publication.

Reply: We thank the reviewer for the helpful comments. See below our replies to the specific comments.

**Specific comments to the paper:**

1. Line 62: the role of the pore connection should be also mentioned, porosity is an important factor in flow and transport as well as the continuity between pores. This notion should appears in the introduction, as well as tortuosity, which then used in the proposed methodology.

Reply: We agree. We revised the text accordingly: *"Furthermore, the connectivity of the different porosities often results in a fracture-cave network, which dominates the flow structures in karst systems (Zhang, 2022)"*. The following was added to the reference list in the revised text: Zhang, X., Huang, Z., Lei, Q., Yao, J., Gong, L., Sun, S., and Li, Y.: Connectivity, permeability and flow channelization in fractured karst reservoirs: A numerical investigation based on a two-dimensional discrete fracture-cave network model, Adv. Water Resour., 161, 104142, https://doi.org/10.1016/j.advwatres.2022.104142, 202

2. Line 104: "well-defined spatial catchment" sounds unclear, are the boundaries well defined with the topography and/or geological setting?

Reply: We revised the text to clarify this point: *"This high-alpine karst system has been thoroughly studied and offers a catchment with a well-defined spatial boundary."*.

3. Line 117 "ansatz" is a German word, this appears several time across the manuscript.

Reply: Ansatz is a word in German, but it has been extensively used in the context of physical and mathematical problems and is well-accepted as an international term in the scientific literature.

4. Line 117: "water parcels" the notation sounds unclear is it like "water bodies" between flow zone and dead zone or is it a quantity of water that should be considered as an analogy with an amount of particle ? This point can be clarified in the paragraph.

Reply: Our definition of parcel, in this context, is the classical definition of a fluid parcel in continuum mechanics (Lagrangian approach). We edited the text to reflect this: *"(i.e., infinitesimal volumes of water)"*.

5. Line 123: "water parcels" and "tracer parcels" sounds unclear, a more detailed explanation might help the reader to understand how the analogy is done in the manuscript.

Reply: We revised the text accordingly to clarify this point (see also comment #4 above): *"The ansatz asserts that the accumulation and release of water parcels in the various volumes in the karst system resemble the accumulation and release of "parcels" of a chemical tracer (i.e., infinitesimal volumes of tracer) over time in a porous medium."*.

6. Fig 1. (c) and (d) does the volumetric discharge considers a constant concentration and does the concentration variation assumes a constant discharge ? In my point of view, discharge and concentration cannot be separated so the two might appears on the graph, then the final time series that the author are using will appears more clear to the reader. At some point, the present version bring some confusion with the widely used tracer BTC.

Reply: We present the analogy of measured discharge and measured tracer concentration separately as each of them can be measured and presented as a time series. The

clarifications made in the revised text for comments #4 and #5 (see above) and #7 (see below) will help clarify this point.

7. Line 172: "C(t) is reinterpreted as translated to a water volume" please clarify the underlying hypothesis to move from concentration curve to fluxes curve.

Reply: We revised the text to clarify this point: *"The concentration at a specific point is analogous to the moisture content, and the classical C(t) breakthrough curve is analogous to the (volumetric) amount of water per time reaching the domain outlet (or measurement plane)."*.

8. Fig 3. I don't think we can consider the fit as satisfactorily. The data do not exhibit a normal distribution shape, also the fitted distribution provides negative distances. The fitted distribution should then at least be a truncated distribution to avoid negative values.

Reply: During development of the simulation, different distributions were examined, and the end results were similar. In our findings, we concluded that a normal distribution describes the statistical distribution of distances in this case quite well. We revised the text to address this: *"A normal distribution was chosen as a simplified representation of the distance distribution; preliminary simulation results were similar for different skewed distributions."*. Note that we do set negative values to zero as stated in the text: "(physically unrealistic, negative sampled values are set to 0)".

9. Fig 5. How is the curvature of the likelihood distribution fixed? Is it a working hypothesis or is it derived from somewhere else?

Reply: This is an example for $SF_I$=0.01% as stated in the text. We edited the revised caption as it was indeed missing and might be confusing for the reader.

10. Line 283: 'optimization is achieved' How is the optimization performed? Which methodology/algorithm? I would recommend also to write "model parameter estimation" rather than "optimization".

Reply: We revised the text accordingly: *"Given the presence of multiple model parameters, optimization is achieved by applying a bound constraint version of the MATLAB fminsearch function (D'Errico, 2024) to minimize the Root Mean Squared Error (RMSE) between observed and simulated discharge. A broad range of constrained parameters were investigated, as detailed in Table 1. The 2016 dataset was first utilized for model parameter estimation, while the 2017 and 2018 datasets served as targets for validation, by considering them for prediction using the optimized parameters from the 2016 dataset."*.

11. Line 293: Why using NSE values that tends to favorize large values while the focus seems to be on tailing? Other performance criteria for model evaluation would be more suitable, or another option would be to compute NSE on a variable transformation such as 1/Q or squared_root(Q)

Reply: While the focus of this study was indeed fitting the tails, this is the first implementation of the CTRW-PT for modelling karst aquifer discharge. As such, we included a performance criterion that is widely used in hydrological studies (and was specifically used for the karst system in question) to support the overall validity of our model. We have added another performance criterion, often used in KarstMod and revised the text accordingly (The BE scores were added to the figure captions.): *"The Nash-Sutcliffe efficiency (NSE) and modified balance error (BE) were calculated for the optimized simulations, as a measure of the goodness of fit. The NSE and BE are the performance criteria utilized, for example, by the widely used KarstMod software (Frank et al., 2021). They are defined as the normalized variant of the mean squared error and the relative bias of the simulated and observed flow durations, respectively:*

$$NSE = 1 - \frac{\sum(x_s(t) - x_o(t))^2}{\sum(x_o(t) - \mu_o)^2} \qquad (12)$$

$$BE = 1 - \left|\frac{\sum(x_o(t) - x_s(t))}{\sum x_o(t)}\right| \qquad (13)$$

_"_. We also revised the text in the methods to clarify to the reader that other criteria exist as well: _"However, it should be noted that the NSE has limitations when there is large variability in the data, and in some cases other performance criteria may be more relevant for different datasets (see Cinkus et al., 2023a for a comparison of different performance criteria)"_. Please note that Cinkus et al., 2023a was missing from reference list and was added to the revised text as well: Cinkus, G., Mazzilli, N., Jourde, H., Wunsch, A., Liesch, T., Ravbar, N., Chen, Z., and Goldscheider, N.: When best is the enemy of good - critical evaluation of performance criteria in hydrological models, Hydrol. Earth Syst. Sci., 27, 2397–2411, https://doi.org/10.5194/hess-27-2397-2023, 2023a.

12. Table 1: Please add the min and max of the investigated parameter space for the model parameter estimation. Also, please give more information about how the optimized value is estimated (see one of the previous comment) + the estimated tortuosity appears pretty high compared with the literature (e.g. reference below) could you discuss a little on that ? Is it a realistic value for the catchment? Jouves, J., Viseur, S., Arfib, B., Baudement, C., Camus, H., Collon, P., Guglielmi, Y., 2017. Speleogenesis, geometry, and topology of caves: A quantitative study of 3D karst conduits. Geomorphology 298, 86–106. https://doi.org/10.1016/j.geomorph.2017.09.019 Collon, P., Bernasconi, D., Vuilleumier, C., Renard, P., 2017. Statistical metrics for the characterization of karst network geometry and topology. Geomorphology 283, 122–142. https://doi.org/10.1016/j.geomorph.2017.01.034

Reply: We investigated a wide range of the fitting parameters, and we have revised table 1 to present the minimum and maximum of the investigated parameter space. An explanation of how the optimization was achieved was also added to the revised text (see comment #10).

We agree that the estimated tortuosity in the system presented may be considered slightly high, but it is still within a reasonable range for a karst system; see, for example, Assari and Mohammadi (Assari, A., Mohammadi, Z. 2017 Assessing flow paths in a karst aquifer based on multiple dye tracing tests using stochastic simulation and the MODFLOW-CFP

code. Hydrogeo. J. 25 (6). 1679-1702), which describes values between 1.1 and 3.9. Indeed, in some of the cases the tortuosity is higher than 1.6 in the references provided by the reviewer. Therefore, we can consider that the karst system described in this study is rather tortuous. Furthermore, in many karst papers such as those noted by the reviewer, tortuosity is calculated for the branch scale, while in our model the tortuosity is a catchment scale factor. In the case of applying the CTRW-PT for other karst systems, the fitting can benefit from a more detailed characterization of the geometry of the entire system. We revised the text to explain this point: *"The optimized tortuosity factor of 1.6 found for the Disnergschroef system is somewhat higher than that found in some cases (~1.2-1.4, e.g., Jouves et al., 2017; Collon et al., 2017), but well within the range (1.1-3.9) reported for karst systems (e.g., Assari and Mohammadi, 2017). The higher value can be attributed to the morphology of the specific system, and also to the fact that while tortuosity is often calculated at the cave branch scale (e.g., Jouves et al., 2017; Collon et al., 2017), the CTRW-PT model uses a catchment scale tortuosity factor. The variability of tortuosity in different karst morphologies should therefore be recognized when considering different modeling scenarios."*. Please note that the two papers suggested by the reviewer and the Assari and Mohammadi (2017) reference were added to the reference list in the revised text.

13. Line 324: Is it correlation coefficient or correlation pic derived from cross correlation function analysis? What is the time lag considered for the correlation coefficient or what is the lag response obtained based on precipitation-discharge cross correlation function?

Reply: The correlation coefficients considered here are without any time lag, as our aim is to show the variation between the three weather stations. They are different for the three stations although they are located a few kms away from one another.

14. Line 328: This sentence is not useful. I recommend to delete.

Reply: We agree and removed this sentence from the revised text.

15. Line 374: As previously mentioned in my comments, other performance criteria would be more suitable to evaluate the predictive performance of the model regarding high flow and low flow periods respectively. Computing NSE by removing the high value is not a suitable justification for the model improvement on low flow period. Indeed, by skipping the high discharge value to compute the NSE you are changing the benchmark of NSE (the mean of observed time series) so comparison of NSE is not straightforward in that case.

Reply: In light of the comment made by the reviewer, we removed the NSE calculation for the low flow data. Furthermore, we omitted this comparison from the manuscript as it does not help to convey the point of the CTRW-PT fitting of the long tails of the data. Regarding the choice to use NSE, see #11 above.

16. Line 377: Are there some evidence of piston effect with temperature and/or conductivity?

Reply: Yes, and in the following paragraph we state that Frank et al (2021) have reported this.

17. Fig 8. Is not very informative as "sensitivity analysis". Including a sensitivity analysis on the model parameter regarding the model performance would be much more informative. Among the model parameters which one are the more sensitive in the model?

Reply: The figure presents a sensitivity analysis on the new parameters introduced to the model in this study. There are many papers that have already studied sensitivity analysis of different CTRW parameters, but none have done so for the new parameters (as they are new). We edited the caption in the revised text to convey the exact parameters shown in the sensitivity analysis: "*__Figure 8.__ Simulation sensitivity to slow and fast particle contributions, based on the 2016 rainfall data. Simulations that compare different $SF_r$*

*values (a), different $SF_r$ values (b) and only fast/slow particles (c), demonstrate the importance of the slow flow for the observed long tails in the discharge data.*".